# Elevation of Lipid Metabolites in Deceased Liver Donors Reflects Graft Suffering

**DOI:** 10.3390/metabo13010117

**Published:** 2023-01-11

**Authors:** Wei-Chen Lee, Ting-Jung Wu, Chih-Hsien Cheng, Yu-Chao Wang, Hao-Chien Hung, Jin-Chiao Lee, Tsung-Han Wu, Hong-Shiue Chou, Chen-Fang Lee, Kun-Ming Chan

**Affiliations:** Division of Liver and Transplantation Surgery, Department of General Surgery, Chang-Gung Memorial Hospital, Chang-Gung University College of Medicine, Taoyuan 33357, Taiwan

**Keywords:** deceased liver, living donor, liver transplantation, metabolomics

## Abstract

Liver transplantation can be performed with deceased or living donor allografts. Deceased liver grafts are donated from brain- or circulation-death patients, and they have usually suffered from a certain degree of damage. Post-transplant graft function and patient survival are closely related to liver allograft recovery. How to define the damage of liver grafts is unclear. A total of 47 liver donors, 23 deceased and 24 living, were enrolled in this study. All deceased donors had suffered from severe brain damage, and six of them had experienced cardio-pulmonary-cerebral resuscitation (CPR). The exploration of liver graft metabolomics was conducted by liquid chromatography coupled with mass spectrometry. Compared with living donor grafts, the deceased liver grafts expressed higher levels of various diacylglycerol, lysophosphatidylcholine, lysophosphatidylethanolamine, oleoylcarnitine and linoleylcarnitine; and lower levels of cardiolipin and phosphatidylcholine. The liver grafts from the donors with CPR had higher levels of cardiolipin, phosphatidic acid, phosphatidylcholine, phatidylethanolamine and amiodarone than the donors without CPR. When focusing on amino acids, the deceased livers had higher levels of histidine, taurine and tryptophan than the living donor livers. In conclusion, the deceased donors had suffered from cardio-circulation instability, and their lipid metabolites were increased. The elevation of lipid metabolites can be employed as an indicator of liver graft suffering.

## 1. Introduction

Liver transplantation is the only effective treatment for patients with acute or chronic liver failure [1,2,3]. Liver transplantation is also the treatment with the best results for hepatocellular carcinoma (HCC) if the tumors are within Milan or certain criteria [4,5,6,7]. Currently, liver transplantation can be performed with deceased or living donor allografts [1,8]. Living donor liver grafts are donated from healthy persons with good preparation, hence, these grafts can be recognized as normal. Nevertheless, deceased liver grafts are donated from brain- or circulation-death patients who have undergone cardiac or circulation resuscitation. Undoubtedly, the liver grafts will have suffered from a certain degree of damage. To our knowledge, post-transplant graft function and patient survival are closely related to the damage and preservation of liver allografts. Defining the damage of deceased liver grafts is important.

Metabolomics that simultaneously detects many metabolites in bio-fluid or tissues has been employed in the fields of oncology and organ transplantation to evaluate pathological statuses [9,10,11]. The exploration of metabolites in liver allografts may enable surgeons to screen donor organ quality, elucidate the effect of ischemia-reperfusion injury, assess the rate of organ recovery post-operatively, and identify prognostic markers of organ rejection and dysfunction [12]. Cortes et al. correlated the metabolites of early graft dysfunction to the pre-transplant metabolomics profiles of donor liver biopsies. They found that liver graft dysfunction was associated with increased levels of bile acids, lysophospholipids, phospholipids, sphinomyelins and histidine metabolism products [13]. Faitot et al. also found that increased levels of lactate and phosphocholine were associated with graft dysfunction in their real-time metabolic profiles of back-table liver biopsies [14]. Currently, machine perfusion is an emerging novel technique to recover suboptimal organs [15,16]. The exploration of metabolites in liver allografts may help to define suboptimal grafts and find the indicators of machine perfusion.

In the literature, liver allografts for metabolomics studies were taken from deceased donors either after brain death or circulation death. We collected the liver tissues from deceased and living donors of liver transplantation to perform our metabolomics study. As liver grafts from living donors are almost normal, the metabolic difference between deceased and living donor allografts can reflect the suffering of deceased liver grafts. Among the decreased donors, we further defined the difference between the deceased donors with or without cardio-pulmonary-cerebral-resuscitation.

## 2. Materials and Methods

### 2.1. Materials and Patients

For deceased liver donation, the age was not limited, and the accepted criteria of liver function were aspartate aminotransferase (AST) and alanine aminotransferase (ALT) within 10 folds of the upper normal limits (AST < 36U/L and ALT < 36U/L in this hospital), and total bilirubin within 5 folds of the upper normal limits (<1.2 mg/dL in this hospital). The exclusion criteria were active uncontrolled infection and liver cirrhosis by imaging. The living donors were required to have normal liver function, and their ages were between 18 and 55 years. The exclusion criteria included hepatitis B, hepatitis C, moderate-to-severe fatty liver and underling comorbidities. In this study, 47 liver graft donors—23 deceased and 24 living—were enrolled. All the living donors signed their informed consent to donate a small piece of liver for this study. After the liver sample was recovered from the donors, a 3 × 5 mm piece was taken from the edge of the liver at the back table to be studied. This study conformed to the ethical guidelines of the 2000 Declaration of Helsinki and was approved by the institutional review board of Chang-Gung Memorial Hospital (IRB No. 201503383A3).

### 2.2. Extraction of Metabolites from Liver Tissues Using Methanol/Water

A 10–15 mg liver tissue sample was homogenized with 300 μL of 50% methanol. An additional 120 μL of 100% methanol was added and vortexed. An amount of 900 μL of methyl tert-butyl ether was added, mixed thoroughly and rested at room temperature for an hour. An amount of 66 μL of water was added, mixed and rested for 10 min. Then, the sample was centrifuged at 12,000 rpm at 4 °C for 30 min. The upper layer was dried by nitrogen evaporator and stored at −80 °C. To the middle layer, 500% *v*/*v* acetonitrile was added and centrifuged at 12,000 rpm at 4 °C for 30 min. The supernatant was recovered and dried by nitrogen evaporator, and stored at −80 °C.

### 2.3. Metabolomic Analysis by Liquid Chromatography Coupled with Mass Spectrometry (LC-MS)

Liquid chromatographic separation was achieved on a 100 mm × 2.1 mm Acquity 1.7 μm C18 column (Waters Corp; Milford, MA, USA) using an ACQUITY TM Ultra Performance Liquid Chromatography system (Waters Corp; Milford, MA, USA). The column was maintained at 45 °C at a flow rate of 0.5 mL/min. Samples were eluted from the LC column using a linear gradient. The gradient started at 40% B and linearly increased to 99% B within 10 min, and then decreased to 40% B at 10.1 min. Mass spectrometry was performed on a Waters Q TOFMS or Agilent Q TOF operated in positive or negative ion mode. The scan range was from 100 to 1700 *m*/*z*. The desolvation gas was set to 900 L/h at a temperature of 550 °C, the cone gas set to 0 l/h and the source temperature set at 120 °C. The capillary voltage and cone voltage were set to 2500 and 25 V, respectively. The MCP detector voltage was set to 2750 V. The Q TOFMS acquisition rate was set at 0.1 s with a 0.02 s interscan delay [12,17,18].

### 2.4. Data Processing

Metabolomic software, MetaboAnalyst, was used for the multivariate data analysis. Accurate masses of features, which showed significant differences between test groups, were searched against the METLIN, HMDB and KEGG databases. Compound prediction was performed using the Metabolite Database and Molecular Formula Generation software.

### 2.5. Metabolite Identification

For the structural identification of target metabolites, standards were operated under identical chromatographic conditions to that of the profiling experiment. MS and MS/MS analyses were performed under the same conditions. MS/MS spectra were collected at 0.1 spectra per second, with a medium isolation window of ~4 *m*/*z*. The collision energy was set from 5 to 35 V. Several metabolites were further confirmed by an ion mobility mass spectrometer under similar chromatographic conditions.

### 2.6. Statistical Analysis

The significance of the differences between means was determined by a paired or unpaired Student‘s *t* test. Survival rates were calculated using the Kaplan–Meier method and compared between groups using the log-rank test. The statistical analyses were all performed with SigmaPlot 14.0 software for Windows (Systat Softwave, Inc., San Jose, CA, USA). *p* < 0.05 was considered statistically significant.

## 3. Results

### 3.1. Characteristics of Deceased and Living Liver Donors

In total, 47 liver donors—23 deceased and 24 living—were enrolled in this study. Among the 24 living liver donors, all were healthy, and their ages were between 18 and 54 years. In contrast, the 23 deceased donors were older than the living donors, and their ages were between 20 and 72 years. All these donors had suffered from severe brain damage, including 11 with traumatic head injury, 8 with cerebral vascular accidence (6 intracerebral hemorrhage and 2 massive cerebral infarction), and 4 with spontaneous subarachnoid hemorrhage due to aneurysm rupture. Six of them experienced cardio-pulmonary-cerebral resuscitation (CPR) for 4–13 min when they arrived at the emergency room. The levels of AST, ALT and total bilirubin in the deceased donors were higher than in the living donors (*p* < 0.001). Even the renal function was worse in the deceased donors than in the living donors (Table 1).

### 3.2. Lipid Difference between Deceased and Living Donor Liver Grafts

To determine the metabolic difference between the liver grafts donated from deceased or living donors, the liver specimens from the 23 deceased liver grafts and 24 living donor liver grafts were taken for a metabolite analysis. For non-targeted metabolite measurements, 1817 metabolites were detected in total. When focused on lipid metabolites, the difference between the deceased and living donor grafts was shown in a principal component analysis (PCA) score plot (Figure 1). The plot showed a clear separation between the deceased and living donor samples. The metabolites in the living donor liver grafts were similar, but they varied in the deceased liver grafts. Figure 2 shows a representative figure of LC-MS for a deceased male donor (a) and a living male donor (b). When focusing on the fold-difference of metabolites, the deceased liver grafts expressed higher levels of various diacylglycerol (DG), lysophosphatidylcholine (lysoPC), lysophosphatidylethanolamine (lysoPE), oleoylcarnitine and linoleylcarnitine, and lower levels of cardiolipin (CL) and phosphatidylcholine (PC) than the living donor liver grafts in positive ion mode (Table 2). In negative ion mode, the deceased liver grafts expressed higher levels of phosphatidic acid (PA), lysoPC, LysoPE, Docosahexaenoic acid and oleic acid than the living donor liver grafts (Table 3).

### 3.3. Lipid Difference between Deceased Livers with or without Cardiac Arrest and Resuscitation

Among the 23 decreased donors, six patients with 4–13 min of CPR and 16 patients without CPR were included in this portion of the study, excluding one patient without a clear history of CPR. When targeted on lipid metabolism, the liver grafts from donors with or without CPR were different and showed in the PCA score plot (Figure 3). The plot showed that lipid metabolites were more various in the deceased liver grafts with a history of CPR than in the grafts without a history of CPR. The liver grafts from the donors with CPR had higher levels of CL, PC, PE and amiodarone than the donors without CPR in positive ion mode (Table 4), and higher levels of CL, PA and PC in negative ion mode (Table 5).

### 3.4. Lipid Difference between Male and Female Living Liver Donor Livers

Male and female patients have a different body composition. We compared the lipid metabolites of the liver from the 10 male and 14 female living donors. The results showed that the female donors had lower levels of PE, PC and PG in positive ion mode (Table 6), and higher levels of cytidine diphosphate diacylglycerol inositol in negative ion mode (Table 7).

### 3.5. Amino Acid Difference between Deceased and Living Donor Livers

When focusing on the metabolic difference of amino acid in the deceased and living donor liver grafts, the results showed that the deceased livers had higher levels of histidine, taurine and tryptophan than the living donor livers, and the living donor livers had a higher level of valine than the deceased livers (Figure 4).

### 3.6. Amino Acid Difference between Deceased Livers with or without Cardiac Arrest and Resuscitation

When focusing on the metabolic difference of amino acid between the deceased donors with or without CPR, the results showed that the liver grafts from the donors with cardiac resuscitation had higher levels of serine, aspartic acid and proline than the liver grafts from donors without cardiac resuscitation (Figure 5).

### 3.7. Post-Transplant Liver Function

After liver transplantation, the highest levels of AST, ALT and total bilirubin within the first post-transplant week were recorded. These liver function tests were compared between the recipients transplanted with deceased liver grafts and living donor grafts. The median (interquartile) level of AST was 1511 (505–2240) U/L in the recipients with deceased liver grafts, compared to 249 (140–523) U/L in the recipients with living donor grafts (*p* < 0.001). The median (interquartile) level of ALT was 421 (205–971) U/L in the recipients with deceased liver grafts, compared to 154 (86–359) U/L in the recipients with living donor grafts (*p* = 0.007). The total bilirubin levels were not different between the recipients with deceased liver grafts and living donor grafts (5.60 (3.43–10.93) mg/dL versus 5.35 (2.28–8.95) mg/dL, *p* = 0.647).

## 4. Discussion

The deceased liver grafts were donated from brain death donors who might have suffered shock and cardiac resuscitation upon arrival at the emergency room. During their stay in intensive care units, the patients might have suffered from brain injury-related cytokine-releasing syndrome and required the support of high levels of inotropic agents. For the deceased liver grafts, some pre-existing, underlying pathologic conditions were unknown until liver donation. The variations in the deceased liver grafts were much higher than the grafts from living donors. In this study, the age of the deceased donors was higher than that of the living donors. The AST, ALT and total bilirubin levels in the deceased donors were higher than in the living donors. After liver transplantation, the peak levels of AST and ALT in the recipients of deceased donor grafts were also much higher than the recipients of living donor grafts. Clearly, the quality of the deceased liver allografts was inferior to the living donor grafts. However, the exact difference between deceased and living liver grafts has not yet been clarified.

The metabolite analysis of the liver allografts may reflect their pathophysiological change and is a way to understand the final stage of the deceased organs. Using the non-targeted metabolite analysis, lipids were the major metabolite change between the deceased and living donor liver grafts in this study. The deceased liver grafts had more than a two-fold increase in various DG, lysoPC, lysoPE and carnitine. PC and PE are the major components of the cell membrane. LysoPC and lysoPE are derived from PC and PE and are released from apoptotic cells. In this study, the elevation of lysoPC and lysoPE reflected cell damage in the liver. In a metabolomics study of cirrhotic liver, McPhail et al. reported that lipids were elevated in their non-survival cirrhotic patients compared with survival cirrhotic patients [19]. Serkora reported a patient with two consecutive liver transplantations due to the failure of the first liver graft, and the metabolite study showed an increase in total fatty acid in the first liver transplantation compared with the second liver transplantation [20]. Cortes et al. divided their liver transplant recipients into an early allograft dysfunction group and an immediate graft function group. They found that liver graft dysfunction was associated with increased levels of bile acids, lysophospholipids, phospholipids, sphinomyelins and histidine in pre-transplant liver allograft biopsies [13]. Obviously, the elevation of lipid metabolites was associated with graft dysfunction. The deceased donors had suffered from cardiac or circulation distress for a period of time, and the liver had unpreventably suffered from hypoperfusion, which resulted in cell damage. Lipid elevation reflected the damage of the liver grafts.

The liver grafts from the deceased donors with CPR would be further damaged compared with the deceased donors without CPR. Therefore, the deceased donors were further divided into the donors with or without CPR to study the metabolites. The lipid metabolite analysis revealed that CL (80:12)/CL (80:0)/CL976:0, PA (18:1/14:0), PC (34:3)/PC (38:3) and PE-NMe (18:1/18:2)/PE-NMe (33:0)/PE-NMe (38:6) were increased by more than 1.5-fold in the deceased donors with CPR compared with the deceased donors without CPR. PA, PC and PE were all components of the cell membrane, and CL was located at the inner membrane of the mitochondria and released from apoptotic cells. The increase in these lipids evidenced that damage of the liver cells was more severe in the deceased donors with CPR than the donors without CPR.

Body composition differs between males and females. The lipid metabolism in the liver may also be different in males and females. In this study, the lipid metabolites from the male and female living liver grafts were compared, revealing that the PG, PE and PC levels were almost seven-fold lower in the female livers than in the male livers. Sexual lipidomic dimorphism is interesting in metabolomics study of clinical diseases [21]. The difference of lipids in female or male livers may be related to rejection or long-term outcomes in liver transplantation. However, the limited data in this study could not define the significance of lipid difference in male or female liver grafts, and further studies are needed.

In this non-targeted metabolite analysis, there were amino acid differences between the deceased and living donor grafts. The deceased liver grafts had higher levels of histidine, taurine and tryptophan and a lower level of valine than the living donor grafts. If the donors had CPR, the serine and proline levels were higher than the grafts from deceased donors without CPR. Amino acids are involved in biosynthesis, the tricarboxylic acid cycle (TCA) cycle and urea cycle metabolism. In this study, different levels of amino acids in the TCA cycle were related to mitochondrial function, which contributes to energy production and synthesis. However, the meanings of these amino acid alterations were not clear. Xu et al. targeted five metabolites in the purine pathway from donations after brain death or circulation death and claimed that a panel composed of purine metabolites and ALT could predict early graft function [22]. Nevertheless, further studies were needed to clarify the meaningful alteration of amino acids.

Machine perfusion is a new technique to perfuse extended criteria grafts [15,16,23]. Some impending discarded liver grafts can be rescued and transplanted with proper liver function. In this study, lipid metabolite elevation of the liver grafts implied their suboptimal quality and might lead to early graft dysfunction. The elevation of lipid metabolites may be employed as an indicator of machine perfusion to preserve organ function, although further study is needed.

## 5. Conclusions 

The quality of liver allografts is closely related to the outcomes of liver transplantation. The deceased donors had suffered from cardio-circulation instability and other conditions before donation. There is no doubt that the quality of deceased liver grafts is inferior to that of living donor grafts. However, it is difficult to judge the quality of deceased liver allografts grossly. Through metabolomics study, lipid metabolites were increased in the deceased allografts compared with the living donor grafts. The elevation of lipid metabolites can be employed as indicators of liver graft suffering.

## Figures and Tables

**Figure 1 metabolites-13-00117-f001:**
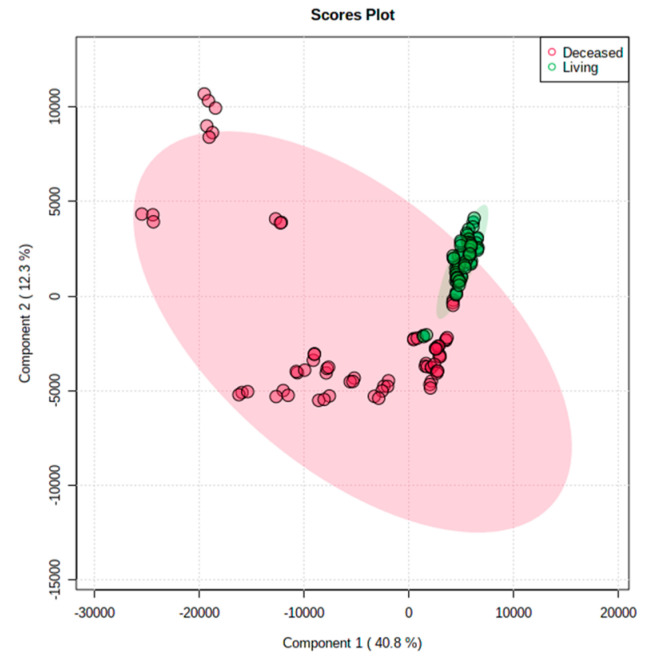
The principal component analysis score plots of deceased and liver donor liver grafts. The plot shows that the metabolites in the living donor liver grafts were similar, but the metabolites varied in the deceased liver grafts.

**Figure 2 metabolites-13-00117-f002:**
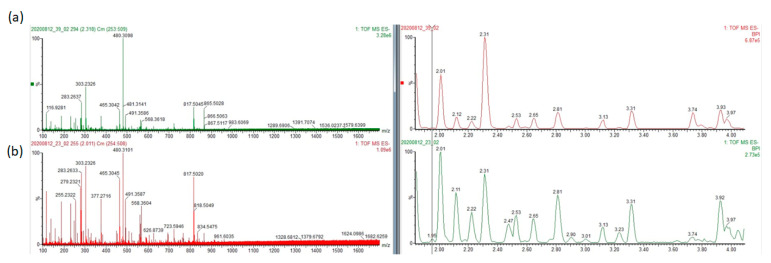
A representative figure of LC-MS between 2 and 4 min for a deceased male donor (**a**) and a living male donor (**b**).

**Figure 3 metabolites-13-00117-f003:**
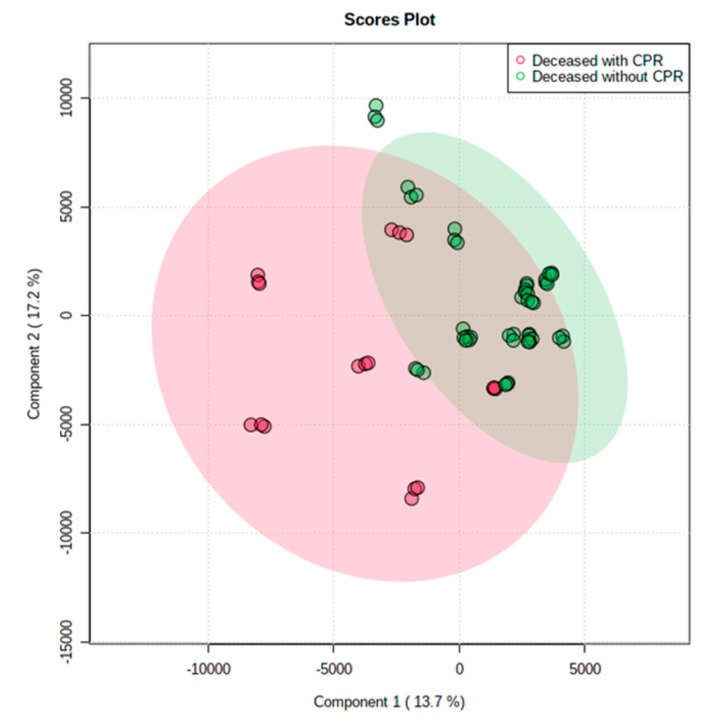
The principal component analysis score plots of deceased liver grafts with or without CPR. The plot showed that lipid metabolites were more various in deceased liver grafts with a history of CPR than without a history of CPR.

**Figure 4 metabolites-13-00117-f004:**
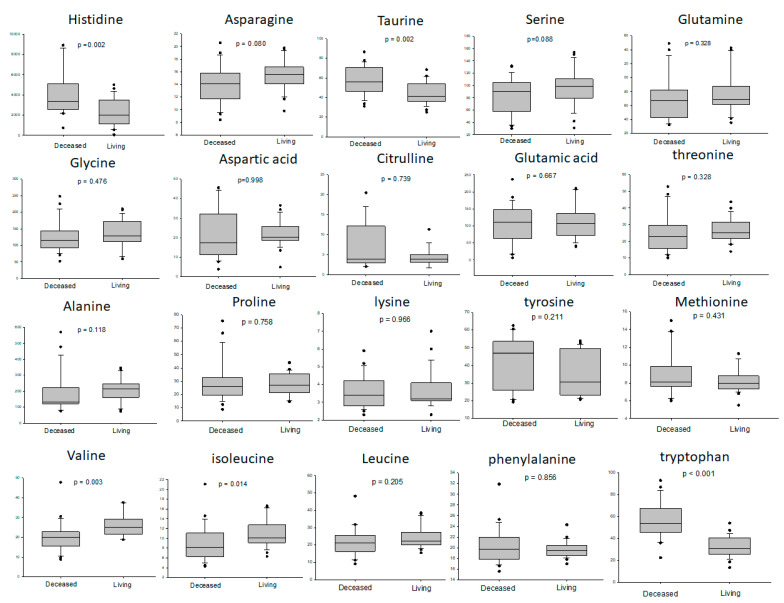
The difference of amino acid in deceased and living donor liver grafts. The deceased livers had higher levels of histidine, taurine and tryptophan than living donor livers, and living donor livers had a higher level of valine than deceased livers.

**Figure 5 metabolites-13-00117-f005:**
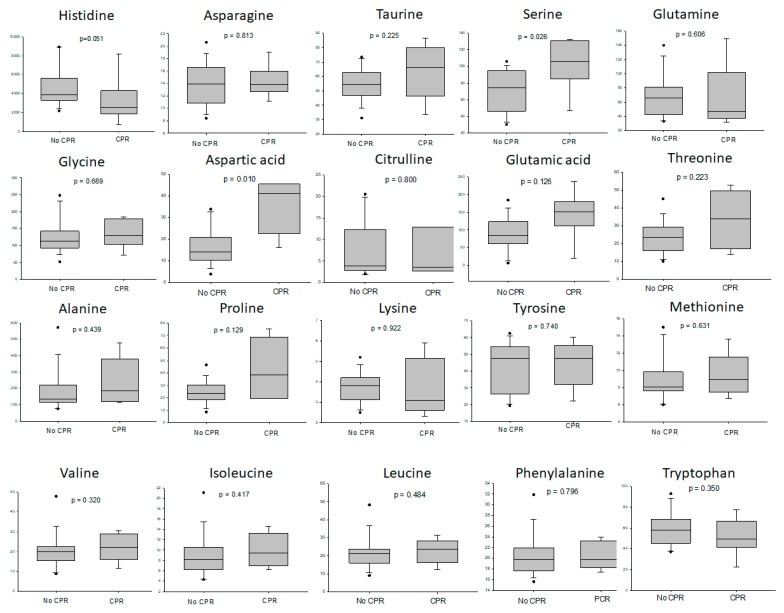
The difference of amino acid between the deceased donors with or without CPR. The liver grafts from the donors with CPR had higher levels of serine, aspartic acid and proline than the liver grafts from donors without CPR.

**Table 1 metabolites-13-00117-t001:** The characteristics of deceased and living liver donors.

	Deceased (*n* = 23) Median (Interquartile) (Range)	Living (*n* = 24) Median (Interquartile) (Range)	*p*
Gender (M/F)	19/4	10/14	0.006
Age (years) Liver function	52 (45–59) (20–72)	31 (22.3–40.5) (18–54)	<0.001
AST (u/L)	50 (32–94) (16–654)	22 (17–24.8) (13–32)	<0.001
ALT (u/L)	39 (20–85) (6–280)	14 (11–19.8) (8–39)	<0.001
T. Bil (mg/dL) Renal function	0.8 (0.6–1.3) (0.4–2.1)	0.45 (0.4–0.5) (0.2–1.8)	<0.001
BUN (mg/dL)	21 (12.5–27.5) (7–72)	13.6 (12.2–17.5) (7.2–24.1)	<0.001
Creatinine (mg/dL)	1.33 (0.79–2.78) (0.38–7.25)	0.64 (0.53–0.90) (0.46–1.38)	<0.001

AST, aspartate aminotransferase; ALT, alanine aminotransferase; T. bil, total bilirubin, BUN, blood urea nitrogen.

**Table 2 metabolites-13-00117-t002:** The difference of lipidomics in MS-electrospray positive ion mode between deceased and living liver donors.

Metabolite ID	Putative ID	*m/z*	Retention Time (Min)	Adduct Ion	VIP Score	Fold Change (Deceased/Living)	*p* Value
Met4565	TG (56:8)	925.726	8.223567	M + Na	6.1092	0.59264	<0.001
Met3932	TG (48:1)	827.7102	8.545533	M + Na	5.4201	1.292	0.021
Met2635	DG (36:2)	643.5283	6.693683	M + Na	13.954	2.7496	<0.001
Met2447	DG (34:2)	615.4963	6.322617	M + Na	13.679	2.0373	<0.001
Met2622	DG (36:3)	641.5123	6.3531	M + Na	13.43	2.6862	<0.001
Met2457	DG (34:1)	617.5124	6.670433	M + Na	11.75	2.7726	<0.001
Met2610	DG (38:7)	639.4973	5.98635	M + H	6.4279	2.4097	<0.001
Met2266	DG (32:1)	589.4811	6.223217	M + H	5.8665	3.1871	<0.001
Met2650	DG (36:1)	645.5435	7.069117	M + H	5.6524	2.8225	<0.001
Met5322	CL (78:2)	1546.088	5.346233	M + H	8.2377	0.37392	<0.001
Met5288	CL (78:11)	1528.041	5.337417	M + NH4	7.0691	0.38442	<0.001
Met5341	CL (79:2)	1560.101	4.983233	M + H	5.3968	0.10088	0.029
Met3117	PC (32:2)	730.5389	4.848617	M + H	7.2412	0.59041	<0.001
Met1942	LysoPC (18:0)	524.3717	2.240733	M + H	5.0108	2.0887	<0.001
Met1832	LysoPC (16:0)	496.3401	1.621633	M + H	8.0022	2.3737	<0.001
Met1778	LysoPE (18:0)	482.3247	2.336933	M + H	6.3474	7.0726	<0.001
Met1677	LysoPE (16:0)	454.2929	1.690217	M + H	5.4592	7.3048	<0.001
Met1567	Oleoylcarnitine	426.3577	1.583533	M + H	6.1608	3.3489	<0.001
Met1555	Linoleyl carnitine	424.3424	1.246367	M + H	5.1404	3.4191	<0.001

**Table 3 metabolites-13-00117-t003:** The difference of lipidomics in MS-electrospray negative ion mode between deceased and living liver donors.

Metabolite ID	Putative ID	*m/z*	Retention Time (Min)	Adduct Ion	VIP Score	Fold Change (Deceased/Living)	*p* Value
Met5575	CL (18:0/16:1/18:2/18:0)	1430.0226	5.52	M − H	5.6907	1.6482	0.0006
Met4042	PC (38:5)	852.5745	5.50	M − H	5.8969	0.56657	<0.001
Met3343	PA (18:1/14:0)	773.5335	5.40	M + FA − H	5.439	1.5023	<0.001
Met1373	PA (20:0)	465.3036	2.81	M − H	5.9074	1.6195	<0.001
Met3796	PE-NMe (38:5)	824.5443	4.96	M + FA − H	6.2031	0.64515	0.007
Met2554	PE (18:1/14:0)	688.4920	5.41	M − H	5.3684	1.6523	<0.001
Met1717	LysoPC (16:0)	540.3302	1.61	M + FA − H	5.466	2.0806	<0.001
Met1631	LysoPE (22:6)	524.2778	1.22	M − H	5.4961	0.40577	<0.001
Met1427	LysoPE (18:1)	478.2931	1.78	M − H	5.9285	8.5482	<0.001
Met1448	LysoPE (18:0)	480.3089	2.32	M − H	18.204	9.0232	<0.001
Met1299	LysoPE (16:0)	452.2774	1.68	M − H	14.75	9.3022	<0.001
Met0805	Docosahexaenoic acid	327.2321	1.84	M − H	16.492	7.9923	<0.001
Met0527	Oleic acid	281.2480	2.65	M − H	16.403	19.672	<0.001

**Table 4 metabolites-13-00117-t004:** The difference of lipidomics in MS-electrospray positive ion mode between deceased donors with or without CPR.

Metabolite ID	Putative ID	*m/z*	Retention Time (Min)	Adduct Ion	VIP Score	Fold Change (CPR/No CPR)	*p* Value
Met4392	TG (54:6)	896.7696	8.193083	M + NH4	5.9231	0.51968	0.009
Met3410	TG (44:1)	771.6465	8.078783	M + Na	5.0154	1.8289	0.042
Met5332	CL (80:12)	1554.047	6.04935	M + H	6.5833	4.1416	0.0006
Met5433	CL (80:0)	1600.126	6.04935	M + Na	5.7671	3.6452	<0.001
Met5318	CL (76:0)	1544.066	5.55215	M + Na	5.3423	2.4026	0.0005
Met3789	PC (38:3)	812.6169	6.0574	M + H	14.103	1.432	0.011
Met3997	PE-NMe (18:1/18:2)	834.5989	6.0574	M + Na	9.7381	1.504	0.002
Met3184	PE-NMe (33:0)	742.5395	5.5921	M + H	8.2295	1.6307	0.0004
Met3460	PE-NMe (38:6)	778.537	5.1485	M + H	5.1207	3.226	<0.001
Met2653	Amiodarone	646.032	1.751167	M + H	8.4211	2.6749	0.012

**Table 5 metabolites-13-00117-t005:** The difference of lipidomics in MS-electrospray negative ion mode between deceased donors with or without CPR.

Metabolite ID	Putative ID	*m/z*	Retention Time (Min)	Adduct Ion	VIP Score	Fold Change (CPR/No CPR)	*p* Value
Met5566	CL (72:8)	1421.9490	7.67	M − H	7.797	1.4559	0.014
Met5795	CL (72:0)	1554.1446	6.03	M + FA − H	5.0505	1.7346	0.0008
Met3343	PA (18:1/14:0)	773.5335	5.40	M + FA − H	6.1291	1.5509	0.013
Met3589	PC (34:3)	800.5441	5.11	M + FA − H	8.8264	1.8179	0.0005

**Table 6 metabolites-13-00117-t006:** The difference of lipidomics in MS-electrospray positive ion mode between female and male living liver donors.

Metabolite ID	Putative ID	*m/z*	Retention Time (Min)	Adduct Ion	VIP Score	Fold Change (Female/Male)	*p* Value
Met3426	PE (36:4)	773.553	4.011383	M + NH4	5.0629	0.14563	<0.001
Met3275	PC (34:3)	756.5534	3.91995	M + H	5.2905	0.15055	0.0136
Met3850	PC (36:2)	818.5923	3.57515	M + H	6.4626	0.14828	<0.001
Met3740	PG (36:1)	808.571	2.233117	M + NH4	8.3152	0.009864	<0.001

**Table 7 metabolites-13-00117-t007:** The difference of lipidomics in MS-electrospray negative ion mode between female and male living liver donors.

Metabolite ID	Putative ID	*m/z*	Retention Time (Min)	Adduct Ion	VIP Score	Fold Change (Female/Male)	*p* Value
Met5123	CDP-DG (37:0)	1068.6853	6.37	M + FA − H	5.488	10.185	0.009
Met4063	Inositol nicotinate	855.1110	5.85	M + FA − H	5.6054	0.53003	0.006
Met3903	PS (36:1)	834.5479	3.05	M + FA − H	5.0324	0.14365	<0.001
Met4041	PE-NMe (44:10)	852.5590	2.21	M − H	5.974	0.44512	<0.001

## Data Availability

All data generated or analyzed during this study are included in this published article.

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
