# Peer review of "Elevation of Lipid Metabolites in Deceased Liver Donors Reflects Graft Suffering"

_metabolites, 2023, doi:10.3390/metabo13010117_

Round 1
Reviewer 1 Report
The authors explored liver graft metabolomics by liquid chromatography coupled with mass spectrometry in this study. 47 liver donors were enrolled in this study to compare the metabolites between living donor grafts and the deceased liver grafts. The authors found the deceased donors were suffered from cardio-circulation un-stability and increased lipid metabolites.
The manuscript can be revised as follows:
Introduction: More background information should be provided, especially the research progress on the metabolic analysis of liver graft. Besides, the novelty of this study should be pointed out.
2.1. Materials and Patients: The age and gender of donors should be given.
L75: What are the HPLC linear gradient conditions used?
For the methods, authors should provide relevant references.
Table 1: BUN should be defined. Moreover, the data should be more clearly presented. Data in Table 1 are very difficult to understand.
Table 2-5: Please note the significant digit of each data.
The quality of Figs. 3 and 4 is low and must be improved.
More discussion is needed.
The conclusion should be extended.
More relevant references should be added.
Author Response
Dear Reviewer:
We are pleased that we have the opportunity to revise our manuscript. We thank your comments and suggestions. Point-to-point responses are described as following:
- Introduction: More background information should be provided, especially the research progress on the metabolic analysis of liver graft. Besides, the novelty of this study should be pointed out.
Reply: We add the some background information in Introduction section.
- Materials and Patients: The age and gender of donors should be given.
Reply: The age and gender of donors are given, and described in Table 1.
- What are the HPLC linear gradient conditions used? For the methods, authors should provide relevant references.
Reply: The conditions are added in Materials and Methods section 2.3. The condition was “The gradient started with 40% B, increased to 99% B linearly within 10 minutes, and deceased to 40% B at 10.1minutes.” The reference was cited.
- Table 1: BUN should be defined. Moreover, the data should be more clearly presented. Data in Table 1 are very difficult to understand.
Reply: BUN was defined as blood urea nitrogen. Table 1 was revised.
- Table 2-5: Please note the significant digit of each data.
Reply: Table 2-5 were revised.
- The quality of Figs. 3 and 4 is low and must be improved.
Reply: Fig 3 and 4 were replaced by Box plots.
- More discussion is needed. The conclusion should be extended.
Replay: Discussion and conclusion were slightly extended.
Thank you for your comments and suggestions again. Hopefully, our manuscript can be accepted.
Sincerely yours,
Wei-Chen Lee
Reviewer 2 Report
Thank you for the opportunity to review this manuscript, dealing with interesting findings entitled “Elevation of lipid metabolites in deceased liver donors reflects graft suffering”. This study includes 47 liver donors, 13 deceased, and 24 living donors’ samples. All the 23 deceased donors suffered 14 from severe brain damage and six of them experienced cardio-pulmonary-cerebral resuscitation 15 (CPR). Exploration of liver graft metabolomics was conducted by liquid chromatography coupled 16 with mass spectrometry. As per liquid chromatography data compared with living donor grafts, the deceased liver grafts expressed 17 higher levels of various diacylglycerol, lysophosphatidylcholine, lysophosphatidylethanolamine, oleoylcarnitine, and linoleylcarnitine; and lower levels of cardiolipin and phosphatidylcholine. Interestingly, they observed that the liver grafts from the donors with CPR had higher levels of cardiolipin, phosphatidic acid, phosphatidylcholine, phatidylethanolamine, and amiodarone than the donors without CPR. Furthermore, the deceased livers had higher levels of histidine, taurine, and tryptophan than living donor livers. As per the observations, they proposed well that deceased donors suffered from cardio-circulation un-stability and lipid metabolites were increased and elevation of lipid metabolites can be employed as indicators of liver graft suffering. The work is well organized and comprehensively described. All cited work is appropriate and related to previous work. All findings are interesting, and the article includes a balanced and critical view of the outcomes. However, a few things/explanations are missing in this article in addition to the lack of good graphic images.
· It would be better if the author can add inclusion and exclusion criteria for the selection of donors. It will clear the conclusion in justification with other diseases or disease states.
· The author needs to add at least one representative image of LCMS for each observation in the article or as supplementary material which will confirm their interpretations and accuracy or estimation.
· Do they observe any significance of data with sex differentiation? The author needs to add a few points of para in the discussion section.
· All tables are good while the images are of poor graphical quality. The author needs to revise all images with good graphics images. It would be nice if the author could replace the bar graph with a scatter bar graph to show the individual value of their observations.
Author Response
Dear Reviewer:
We are pleased that we have the opportunity to revise our manuscript. We thank your comments and suggestions. Point-to-point responses are described as following:
- It would be better if the author can add inclusion and exclusion criteria for the selection of donors. It will clear the conclusion in justification with other diseases or disease states.
Reply: The inclusion and exclusion criteria of donors were added in Materials and Methods section 2.1
- The author needs to add at least one representative image of LCMS for each observation in the article or as supplementary material which will confirm their interpretations and accuracy or estimation.
Reply: A representative figure of LC-MS was added as Fig. 2
- Do they observe any significance of data with sex differentiation? The author needs to add a few points of para in the discussion section.
Reply: The significance of data with sex differentiation was added as Table 6 and 7.
- All tables are good while the images are of poor graphical quality. The author needs to revise all images with good graphics images. It would be nice if the author could replace the bar graph with a scatter bar graph to show the individual value of their observations.
Reply: Fig 3 and 4 were replaced by Box plots.
Thank you for your comments and suggestions again. Hopefully, our manuscript can be accepted.
Sincerely yours,
Wei-Chen Lee